

# A model study of warming-induced phosphorus-oxygen feedbacks in open-ocean oxygen minimum zones on millennial timescales

Daniela Niemeyer[1], Tronje P. Kemena[1], Katrin J. Meissner[2], Andreas Oschlies[1]

[1]Helmholtz-Zentrum für Ozeanforschung Kiel (GEOMAR), Düsternbrooker Weg 20, 24105 Kiel, Germany
[2]Climate Change Research Centre and ARC Centre of Excellence for Climate System Science, University of New South Wales, Level 4 Mathews Building, Sydney, New South Wales, 2052, Australia

*Correspondence to*: D. Niemeyer (dniemeyer@geomar.de)

**Abstract.** Observations indicate an expansion of oxygen minimum zones (OMZs) over the past 50 years, likely related to ongoing deoxygenation caused by reduced solubility, changes in stratification and circulation, and a potential acceleration of
organic matter turnover in a warming climate. Higher temperatures also lead to enhanced weathering on land, which, in turn, increase the phosphorus and alkalinity flux into the ocean. The overall area of ocean sediments that are in direct contact with low oxygen bottom waters also increases with expanding OMZs. This leads to an additional release of phosphorus from ocean sediments and therefore raises the ocean's phosphorus inventory even further. Higher availability in phosphorus enhances biological production, remineralisation and oxygen consumption, and might therefore lead to further expansions of
OMZs, representing a positive feedback. A negative feedback arises from the enhanced productivity-induced drawdown of carbon and also increased uptake of $CO_2$ due to increased alkalinity, which, in turn, got there through weathering. This feedback leads to a decrease in atmospheric $CO_2$ and weathering rates. Here we quantify these two competing feedbacks on millennial timescales for a high $CO_2$ emission scenario. Using the UVic Earth System Climate Model of intermediate complexity, our model results suggest that the positive benthic phosphorus release feedback has only a minor impact on the
size of OMZs in the next 1000 years, although previous studies assume that the phosphorus release feedback was the main factor for anoxic conditions during Cretaceous period. The increase in the marine phosphorus inventory under assumed business-as-usual global warming conditions originates, on millennial timescales, almost exclusively from the input via terrestrial weathering and causes a 4 to 5-fold expansion of the suboxic water volume in the model.



## 1 Introduction

Oxygen minimum zones (OMZs) have been expanding over the past 50 years and it has been suggested that this expansion is related to recent climate change (STRAMMA ET AL., 2008), though current $CO_2$ emission-forced models are challenged to reproduce this expansion in detail (STRAMMA ET AL., 2012; CABRE ET AL., 2015). There are at least three different processes

that can have an impact on the size of OMZs in a warming climate: Ocean warming and its impact on solubility of $O_2$ in the ocean (BOPP ET AL., 2002), changes in ocean dynamics, e.g. stratification, convective mixing and circulation (MANABE & STOUFFER, 1993; SARMIENTO ET AL., 1998), biological production effects (BOPP ET AL., 2002) including possible $CO_2$-driven changes in stoichiometry (OSCHLIES ET AL., 2008) and $CO_2$ induced changes in ballasting particle export (HOFMANN & SCHELLNHUBER, 2010). Here we investigate how changes in biological production and subsequent remineralisation can

affect OMZs in addition to the above-mentioned thermal and dynamic effects. We focus on changes in the phosphorus (P) cycle. P is the main limiting nutrient on long timescales (PALASTANGA ET AL., 2011) and we examine possible effects of changes in the P cycle on millennial timescales.

The major source of P for the ocean is the river input (FILIPPELLI, 2008; PAYTON & MCLOUGHLIN, 2007; FÖLLMI, 1996, PALASTANGA ET AL., 2011; FROELICH ET AL., 1982), which is determined by terrestrial weathering of apatite (FILIPPELLI,

2002; FÖLLMI, 1996). The main factors controlling terrestrial weathering are temperature, precipitation and vegetation. Higher temperatures are generally associated with enhanced precipitation and in many places with higher terrestrial net primary productivity (MONTEIRO ET AL., 2012), which all tend to increase weathering rates (BERNER, 1991).

It is difficult to determine how much of the globally weathered P enters the ocean in a bioavailable form. About $0.09 - 0.15$ Tmol $a^{-1}$ of potentially bioavailable P is transported globally by rivers including dissolved organic and inorganic P,

particulate organic P and ironbound P (COMPTON ET AL., 2000). About 25% of this potentially bioavailable P (sum of chemically weathered flux, shale-derived particulate organic P flux and 20% of eolian flux) is trapped in coastal estuaries and will not enter the open ocean (COMPTON ET AL., 2000).

Marine organisms take up P most easily as dissolved inorganic P (DIP). Riverine measurements suggest that only a small fraction of the total P (0.012 to 0.032 Tmol $a^{-1}$) enters the ocean as DIP (FILIPPELLI, 2002; HARRISON ET AL., 2005;

COMPTON ET AL., 2000; WALLMANN, 2010; PALASTANGA ET AL., 2011). But during the passage of estuaries the fraction of DIP could increase by 50% (FROELICH, 1984) to 80% (BERNER & RAO, 1994).

After taking up the bioavailable P for photosynthesis, a large fraction of the newly produced organic matter is exported out of the euphotic zone as detritus (6.42 Tmol P $a^{-1}$ according to the model study by PALASTANGA ET AL., 2011) and the vast majority of this exported organic matter is remineralised in the deeper ocean by bacteria (6.26 Tmol P $a^{-1}$; PALASTANGA ET

AL., 2011), which is an oxygen consuming process. A small fraction of the exported organic matter is deposited at the sediment surface (0.16 Tmol P $a^{-1}$; PALASTANGA ET AL., 2011), about 20% of the deposited P is buried in the sediments on long time scales (0.032 Tmol P $a^{-1}$; PALASTANGA ET AL., 2011), whereas the remaining 80% (0.13 Tmol P $a^{-1}$; PALASTANGA



ET AL., 2011) is released back into the water column as DIP, where it is again available for the uptake of marine primary producers (PALASTANGA ET AL., 2011; WALLMANN, 2010).

The processes of burial and release of P are both redox-dependent. Under oxic conditions the burial rate is high, while under suboxic conditions the benthic release of P is elevated (INGALL & JAHNKE, 1994; KRAAL ET AL., 2012; WALLMANN, 2010;

SLOMP & VAN CAPPELLEN, 2007; FLOEGEL ET AL., 2011; LENTON & WATSON, 2000; TSANDEV, 2010). The redox-dependent release of P into the water column and the decrease in marine oxygen due to remineralisation therefore represent a positive feedback loop on marine biological production (see Fig. 1).

The enhanced detritus export into the ocean interior results in an increased marine uptake of atmospheric $CO_2$. Surface air temperatures decrease with decreasing atmospheric $CO_2$ concentrations, which, in turn, leads to lower weathering rates (see

Fig. 1).

These redox-dependent benthic P fluxes have been investigated in previous studies with the HAMOCC global ocean biogeochemistry model (PALASTANGA ET AL., 2011). PALASTANGA ET AL. (2011) show that doubling the input of dissolved P from rivers results in an increased benthic release of P in their simulations. This leads to a rise in primary production as well as in oxygen consumption, which in turn affects the oxygen availability in sediments. The benthic release of P acts therefore

as a positive feedback on expanding oxygen minimum zones on timescales of 10,000 to 100,000 years (PALASTANGA ET AL., 2011).

Other studies on OMZs focused on the geological past, especially the mid-Cretaceous warm period (120-80 Ma ago) (TSANDEV, 2010; HANDOH & LENTON, 2003; BJERRUM ET AL., 2006; FÖLLMI ET AL., 1996), for which several periods of oceanic oxygen depletion have been inferred from sediment data of black shales (SCHLANGER & JENKYNS, 1976). For

example, the Cretaceous oceanic anoxic event 2 (OAE) at the Cenomanian-Turonian boundary (93.5 Myrs) might represent similar characteristics as the greenhouse world that anthropogenic $CO_2$ emissions may lead us to. Whether similar processes such as surface warming, sea-level rise (HANDOH & LENTON, 2003), and possibly a slow-down of the ocean overturning circulation and vertical mixing (MONTEIRO ET AL., 2012; TSANDEV, 2010) will lead to widespread oxygen depletion in the future, is a reason of concern. Consequently, a better understanding of biogeochemical processes associated with Cretaceous

OAE might help assess the risk of possible future events of low marine oxygen concentrations (TSANDEV, 2010).

In contrast to previous studies with focus on the geological past, we investigate possible future changes over the next 1000 years using an Earth System Climate Model to investigate the feedbacks between the P cycle and OMZs under the Representative Concentration Pathways-Scenario 8.5 (RCP 8.5) of the Intergovernmental Panel on Climate Change (IPCC) AR5 report. The RCP 8.5 scenario is characterized by an increase in atmospheric $CO_2$ concentrations and associated with an

increase in radiative forcing by 8.5 W m$^{-2}$ until year 2100 (in comparison to preindustrial conditions) and is also known as the "business as usual" scenario (RIAHI ET AL., 2011).



## 2 Methods

### 2.1 UVic Model

The UVic Earth System Model (UVic ESCM) version 2.9 (WEAVER ET AL., 2001) is a model of intermediate complexity and consists of a terrestrial model based on TRIFFID and MOSES (MEISSNER ET AL., 2003) including DIC and alkalinity

weathering (MEISSNER ET AL., 2012), an atmospheric energy-moisture-balance model (FANNING & WEAVER, 1996), a $CaCO_3$-sediment model (ARCHER, 1996), a sea-ice model (SEMTNER, 1976; HIBLER, 1979; HUNKE & DUKOWICZ, 1997) and a three-dimensional ocean circulation model (MOM2) (PACANOWSKI, 1995). The ocean model includes a marine ecosystem model based on a nutrient-phytoplankton-zooplankton-detritus model (KELLER ET AL., 2012). The horizontal resolution of all model components is 1.8° latitude x 3.6° longitude. The ocean model has 19 layers with layer thicknesses ranging from 50 m

at the sea surface to 500 m in the deep ocean, it also includes a sub-grid scale bathymetry for benthic processes (SOMES ET AL., 2013).

### 2.2 Phosphorus Cycle in UVic Model

Earlier applications of the UVic ESCM assumed a fixed marine P inventory. We included a representation of the dynamic P cycle for this study. It consists of a modified terrestrial weathering module (MEISSNER ET AL., 2012) and a redox-sensitive

transfer-function for burial (sink) and benthic release (source) of P (FLOEGEL ET AL., 2011; WALLMANN, 2010).

The continental-weathering module developed earlier for fluxes of dissolved inorganic carbon (DIC) and alkalinity (MEISSNER ET AL., 2012; LENTON & BRITTON, 2006), is based on following equations:

$$F_{DIC,w} = F_{DIC,w,0} * \lfloor f_{Si} + f_{Ca} * \left(\frac{NPP}{NPP_0}\right) * \left(1 + 0.087 * (SAT - SAT_0)\right)\rfloor = F_{DIC,w,0} * f\,(NPP, SAT) \qquad (1)$$


$$F_{Alk,w} = F_{Alk,w,0} * \left(\frac{NPP}{NPP_0}\right) * \lfloor f_{Si} * (1 + 0.038) * (SAT - SAT_0) * 0.65^{0.09} * (SAT - SAT_0)\rfloor + f_{Ca} * \left(1 + 0.087 * (SAT - SAT_0)\right) \quad (2)$$

Where $F_{DIC,w}$ and $F_{Alk,w}$ represent the globally integrated flux of DIC and alkalinity via river runoff, $f_{Si}$ and $f_{Ca}$ stand for the fraction of silicate (0.25) and carbonate (0.75) weathering, and $NPP$ and $SAT$ are the global mean net primary production on

land and global mean surface air temperature (in degrees Celsius). The index 0 stands for preindustrial values.

We added the following flux to account for P weathering ($F_{DP,w}$) with the same dependencies on globally and annually averaged net primary production (NPP) and surface air temperature (SAT) as those for DIC:

$$F_{DP,w} = F_{DP,0} * f\,(NPP, SAT) \qquad (3)$$






The global river input of dissolved inorganic P (DIP) is the only continental source for P in the model. The global DIP input is distributed over all coastal points of discharge scaled according to their individual volume discharge. The pre-industrial DIP input to the ocean ($F_{DP,0}$) is assumed to be in steady state and in equilibrium with the total globally integrated burial of P ($BUR_P$):

$$F_{DP,0} = BUR_{P,0} \qquad (4)$$

We use an empirical transfer function for $BUR_P$ and for the benthic release of DIP ($BEN_{DIP}$) derived from observations across bottom-water oxygen gradients (WALLMANN, 2010; FLÖGEL ET AL., 2011). The release of dissolved inorganic P ($BEN_{DIP}$) is calculated as follows:

$$BEN_{DIP} = \frac{BEN_{DIC}}{r_{reg}} \qquad (5)$$

Benthic release of dissolved inorganic carbon ($BEN_{DIC}$) is calculated from an empirical transfer function (Fig. 2 in FLÖGEL ET AL., 2011) to determine $BEN_{DIP}$ fluxes at the ocean bottom. In our model configuration POC is remineralised completely at the ocean bottom and no ocean-to-sediment-fluxes of POC occur, i.e. $BEN_{DIC} = RR_{POC}$, where $RR_{POC}$ denotes the rain rate of particulate organic carbon to the sediment. WALLMANN (2010) calculated $r_{reg}$ by a regression of observational data to bottom-water oxygen concentrations:

$$r_{reg} = \frac{RR_{POC}}{BEN_{DIP}} = Y_F + A * \exp\left(\frac{-[O_2]}{r}\right) \qquad (6)$$

The regeneration ratio is calculated by dividing the depth-integrated rate of organic matter degradation in surface sediments ($RR_{POC}$) by the benthic flux of dissolved P into the bottom water ($BEN_{DIP}$). Parameters are defined as $Y_F = 123 \pm 24$, $A = -112 \pm 24$ and $r = 32 \pm 19$ and $O_2$ is in µmol/l (WALLMANN, 2010). Under oxic conditions $r_{reg}$ is higher than the Redfield ratio (106; REDFIELD ET AL., 1963) and under oxygen-depleted conditions $r_{reg}$ reduces to 10 (WALLMANN, 2010).

The rain rate of POP ($RR_{POP}$) is calculated by the rain rate of POC ($RR_{POC}$) divided by the Redfield Ratio. As a result $BUR_P$ can be calculated as follows:

$$BUR_P = RR_{POP} - BEN_{DIP} \qquad (7)$$

The burial of P ($BUR_P$) in the sediment is equal to the rain rate of particulate organic P ($RR_{POP}$) minus $BEN_{DIP}$ (FLOEGEL ET AL., 2011).





### 2.3 Model Simulations

Two model simulations were performed. Our control simulation, called simulation REF hereafter, neither includes weathering nor benthic fluxes of P. The global amount of P in the ocean is therefore conserved in this simulation over time.

The second simulation, called WB, includes both, P weathering and benthic fluxes of P. The spin up has performed by computing the benthic fluxes according to Eq. 6. The weathering fluxes were set to a value to compensate the burial rate at each time step (Eq. 4) during the spin up but not thereafter.

After a spin-up of 20,000 years under preindustrial boundary conditions, we forced the model with anthropogenic $CO_2$ emissions following the RCP 8.5 scenario of the IPCC AR5 assessment (MEINSHAUSEN ET AL., 2011; RIAHI ET AL., 2011).

The $CO_2$ emissions in the UVic ESCM reach 105.6 Pg $CO_2$ $a^{-1}$ in year 2100. Between years 2100 and 2150 the models are forced with constant $CO_2$ emissions (105 Pg $CO_2$ $a^{-1}$), followed by a linear decline until year 2250 to a level of 11.5 Pg $CO_2$ $a^{-1}$ and then linearly to zero emissions in year 3005 (see Fig. 2a). Simulated atmospheric $CO_2$ concentrations peak in year 2250 with 2148.6 ppmv and equal 1835.8 ppmv in year 3005 (see Fig. 2a).

### 2.4 Simulated preindustrial equilibrium

The UVic ESCM has been validated under present day and preindustrial conditions in numerous studies (EBY ET AL., 2009; WEAVER ET AL., 2001). In particular, KELLER ET AL. (2012) recently compared results of its ocean biogeochemical component to observations and previous model formulations. We therefore concentrate our validation on the new model component in this study, the P cycle.

Estimates of burial rates vary over a wide range in the literature. PALASTANGA ET AL. (2011) report a value of 0.032 Tol P $a^{-1}$

based on simulations with the HAMOCC-model, while FILIPPELLI (2002) suggests a range between 0.065 and 0.097 Tmol P $a^{-1}$ based on observations. WALLMANN (2014) calculated a much higher burial rate of 0.216 Tmol P $a^{-1}$ with his box model and BATURIN (2007) suggests a burial rate of 0.419 Tmol P $a^{-1}$ based on literature data retrieved at >100 previously unpublished Russian field stations as described in detail by WALLMANN (2010). The burial rate diagnosed by the UVic ESCM under preindustrial boundary conditions (0.38 Tmol P $a^{-1}$), is within range of these earlier estimates.

To conserve marine P during long model spin ups, the dissolved weathering flux of P under preindustrial conditions is set equal to the diagnosed burial rate during the spin-up, 0.38 Tmol P $a^{-1}$. This results in a flux that is by one magnitude higher than estimates based on measurements of DIP (e.g. FILIPPELLI, 2002 (0.032 Tmol P $a^{-1}$) or WALLMANN, 2010 (0.03 Tmol P $a^{-1}$)). However, BERNER & RAO (1994) show that bioavailable P from rivers (P which is either biotically available or associated with biotically related components (PAYTON & MCLAUGHLIN, 2007)) has been underestimated by up to a factor 3.

Due to coastal processes, these fluxes might therefore be much larger than indicated by DIP measurements from river sites. In addition, the potentially reactive P (P that can be transformed to reactive P) flux is thought to be much higher than DIP measurements suggest (COMPTON ET AL., 2000).





Global values for benthic release under preindustrial conditions equals 0.78 Tmol P a$^{-1}$ in the UVic ESCM (simulation WB) while PALASTANGA ET AL. (2011) calculated a global value of 0.13 Tmol P a$^{-1}$ within the HAMOCC model. Based on pore water measurements COLMANN & HOLLAND (2000) estimated a benthic release of 0.84 Tmol P a$^{-1}$ for coastal regions and 0.41 Tmol P a$^{-1}$ for continental slopes.

**3 Results**

**3.1 Simulated Climate**

The global mean atmospheric surface temperature, as simulated by the WB run, increases until year 2835 and peaks at 23.1°C, i.e. 9.91°C above pre-industrial levels. Simulation REF shows similar changes in temperature with an increase until year 2855 and a peak at 23.3°C (see Fig. 2a). Both simulations show a slight recovery in temperatures after the peak (REF:

23.2°C; WB: 23.1°C; year 3005). Atmospheric temperatures in the WB-simulation are slightly lower than in the reference simulation, due to slightly lower carbon dioxide concentrations in the atmosphere, caused by increased alkalinity (REF: 2.498 mol m$^{-3}$; WB: 2.481 mol m$^{-3}$; both for year 3005), the enhanced biological pump and a rise in detritus export rate (see Sect. 3.2), and therefore increased marine uptake of atmospheric $CO_2$. The impact of the negative feedback and its productivity and chemically induced marine uptake of atmospheric $CO_2$ on surface air temperatures is thus small compared

to the $CO_2$ induced warming in a high-emission scenario.

This negative feedback also results in a larger marine carbon inventory in the WB-simulation and a somewhat subdued expansion of OMZs (see Sect. 3.3).

Given that the response in temperature is similar for both simulations, differences in oxygen concentration mainly originate from biogeochemical changes, which will be discussed below.

**3.2 Phosphorus Dynamics**

The weathering rate (see Fig. 3b) and associated flux of P into the ocean via river discharge more than doubles relative to the pre-industrial situation in our WB-simulation and leads to an enhancement in global mean oceanic P concentrations by 27% over 1000 years (see Fig. 2b). At the same time, enhanced benthic burial acts as a P sink, mitigating the total increase in marine P. The P concentration remains constant in the control run REF.

The weathering input in the WB-simulation is largest north of 30°N (0.338 Tmol P a$^{-1}$ in year 3005; see Fig. 3a), while south of 30°S (0.138 Tmol P a$^{-1}$) and in the low latitude Pacific Ocean the input is lowest (0.117 Tmol P a$^{-1}$). Weathering fluxes into the low latitude Indian and Atlantic Oceans equal 0.187 and 0.267 Tmol P a$^{-1}$, respectively.

Increasing P concentrations as well as climate warming result in an increase in net primary production in the ocean (ONPP). Globally integrated ONPP ranges between 43.8 Tmol P a$^{-1}$ (REF) and 44.1 Tmol P a$^{-1}$ (WB) under preindustrial conditions

and 65 Tmol P a$^{-1}$ (REF) and 116.4 Tmol P a$^{-1}$ (WB) at year 3005. The main areas of ONPP increase are located in the tropical ocean, where higher temperatures favour net primary production in the model (results not shown).



Due to enhanced ONPP, the WB-simulation also has a higher export rate (8.6 Tmol P a⁻¹, computed at 130m depth; see Fig. 2b) when compared to the reference run (5.5 Tmol P a⁻¹) in year 3005. In the REF simulation, the export rate declines until year 2175 (4.8 Tmol P a⁻¹) in response to enhanced stratification and associated declining nutrient supply (SCHMITTNER ET AL., 2008; STEINACHER ET AL., 2010; BOPP ET AL., 2013; MOORE ET AL., 2013; YOOL ET AL., 2013). The export rate recovers
to reach 5.5 Tmol P a⁻¹ at the end of the simulation in experiment REF.

The globally integrated remineralisation rate (results not shown) ranges between 5.1 Tmol P a⁻¹ (WB) and 5.2 Tmol P a⁻¹ (REF) in year 1775. Simulation WB is characterized by a strong increase in remineralisation until 3005 with a maximum of 8.1 Tmol P a⁻¹ (in year 3005), while in the reference run the remineralisation rate first decreases, followed by a moderate increase to 5.3 Tmol P a⁻¹. Regions with highest remineralisation are located on the continental margins, especially in the
Indian Ocean.

The P burial in the WB-simulation (results not shown) equals 0.38 Tmol P a⁻¹ in year 1775 and decreases until year 3005 by 44.3% to 0.2 Tmol P a⁻¹. One reason for this decrease is the redox-state of the bottom water. The strong expansion of the suboxic bottom water (see Fig. 2d) in the WB simulation leads to a decrease in benthic burial of P despite an increase in the rain rate of particulate organic P, $RR_{POP}$. In general, burial rates are largest along the coastal margins. Highest increases in
burial rates between years 1775 and 3005 are located in the Arctic Ocean (see Fig. 4a), whereas burial rates decrease in the Bay of Bengal and the Gulf of Mexico where low-oxygen bottom waters expand (see Fig. 5).

The benthic P release (results not shown) increases by 119% until year 3005 to 1.7 Tmol P a⁻¹ (simulation WB). As mentioned above, the benthic release is also a redox-dependent process. This means that an increase in suboxic bottom water area (see Fig. 2d) leads to an enhanced release of benthic phosphate in WB. A rapid increase between years 1775 and 3005
can be found in the Bay of Bengal, the Gulf of Mexico and in the Arctic Ocean (see Fig. 4b).

In our model simulations, both the weathering-induced P flux into the ocean (see Fig. 2c) as well as the net P released from the sediments (see Fig. 2c) show a strong increase under continued global warming, which explains the increase in the marine P inventory in the WB simulation (see Fig. 2b). However, the simulated increase in the weathering input has a much stronger (about 4 times larger) impact on the P budget and therefore on the expansion of OMZs than the benthic release
feedback (see Fig. 2c). We note that even at the end of the thousand-year simulation, the P cycle has not yet reached a new steady state in experiment WB. Weathering rates are high in the warm climate and burial of P has not increased enough to counteract the supply by weathering (see Fig. 3b). The unlimited release of P from sediments also adds to this imbalance. As a result, the marine P inventory is still increasing almost linearly at the end of our simulation. Extending the simulation until year 10,000 reveals that the ocean does not become anoxic (see Sect. 3.3) while the P cycle still exhibits a strong imbalance
between sources and sinks.

### 3.3 Oxygen Response

The black contours in Fig. 5 indicate the lateral extent of OMZs at depths between 300 and 900 m. In year 1775, the suboxic volume, defined here as waters with oxygen concentrations of less than 5 mmol m⁻³, equals 3.9x10⁶ km³ in both simulations



for a depth between 300 and 900 m (see Fig. 2d). An observational estimate of today's suboxic water volume equals $102 \times 10^6$ $\pm\ 15 \times 10^6$ km$^3$ for oxygen concentrations less than 20 mmol m$^{-3}$ (PAULMIER & RUIZ-PINO, 2009), which is considerably larger than the volume of $O_2 < 20$ mmol m$^{-3}$ waters in our simulations ($15.8 \times 10^6$ km$^3$). Comparing our results with observational data from the World Ocean Atlas (WOA), a generally good agreement can be found with regard to spatial

distribution (see Fig. 5). The suboxic areas are located in the upwelling regions of the Eastern Pacific and the Eastern Atlantic as well as in the Indian Ocean (see Fig. 5; representative for both simulations in 1775).

During our transient simulations, we find a considerable expansion of OMZs until year 3005 in both simulations (see Fig. 2d and Fig. 5). The expansion of the suboxic volume between 300 and 900 m is particularly pronounced in the WB simulation where the OMZs account for $4.85 \times 10^7$ km$^3$ between 300 and 900 m depth in year 3005, i.e. an increase by a factor 12.4. The

control simulation (REF) shows a much smaller increase in the volume of OMZs ($1.12 \times 10^7$ km$^3$ between 300 and 900 m depth). As both simulations display similar climates (see Fig. 2a), the difference in the oxygen fields is largely due to the differences in the simulated P cycle.

The sea-floor area in contact with suboxic bottom waters, which directly impacts the redox-sensitive benthic burial and P release, shows an increase by more than a factor of 19 (WB$_{1775}=3.59 \times 10^5$ km$^2$; WB$_{3005}=6.95 \times 10^6$ km$^2$) in the WB simulation

(see Fig. 2d) compared to a factor of 4 increase in the REF simulation (REF$_{1775}=2.79 \times 10^5$ km$^2$; REF$_{3005}=1.2 \times 10^6$ km$^2$).

Despite the substantial expansion of OMZs, the positive feedback involving redox-sensitive release of P from sediments is not strong enough to lead to a run-away feedback leading to large-scale ocean anoxia on millennial timescales. Although the simulated weathering input of P is at the high end of observational estimates, the simulated ocean does not become anoxic, as has been suggested during the Cretaceous. In contrast to the box-model studies of TSANDEV (2011) for the Cretaceous,

that were conducted over one million years, an increase in continental weathering does not result in an OAE under current topography and seawater chemistry - at least not until year 10,000. This suggests that the positive feedback between the release of benthic P and marine net primary production is - in this study and for present day bathymetry - not the decisive factor for a rapid transition into an anoxic ocean.

## 4 Uncertainties

Although the model's subcomponents for weathering, burial and benthic release rates are highly simplified in this study, comparison with observations has shown that the global simulated P fluxes fall within the range suggested by earlier studies (PALASTANGA ET AL., 2011; FILIPPELLI, 2002; BATURIN, 2007; WALLMANN, 2010). The weathering fluxes are calibrated against global mean burial rates under an implicit steady-state assumption, although it is up to now unclear whether the P cycle in the present day ocean is in equilibrium (WALLMANN, 2010). The relatively high P weathering fluxes as well as the

assumed unlimited P reservoir in the sediments in our simulations might lead to an overestimation of the effects on the P cycle and OMZs.





Another uncertainty is the skill of a coarse resolution model to represent continental shelves and coastal areas. Given the importance of these regions for the processes analysed in this study, a high resolution and more sophisticated sediment model should be used in future studies.

It should also be noted that we used present-day bathymetry in this study. Under Cretaceous conditions the impact of the

5 benthic release feedback on OMZs could be very different, and might have been more efficient because the shelf area was considerably larger due to higher sea levels in the Cretaceous (late Cretaceous shelf area: $46 \times 10^6$ km$^2$; present day shelf area: $26 \times 10^6$ km$^2$ (BJERRUM ET AL., 2006)). This expansion might lead to a strengthening of the benthic release feedback, which was shown in the studies of TSANDEV (2011).

In general, we can assume that observations of P sinks are less uncertain than observations of P sources because the results

10 of the P input have a strong dependence on small-scale physical and biological processes e.g. temperature, precipitation as well as lithology, relief, soil and biota (COMPTON ET AL., 2000; FROEHLICH ET AL., 1982), which are subjected to changes on small time and space scales. Changes in marine conditions generally proceed on longer time scales and may simplify the estimations of marine P fluxes.

FILIPPELLI (2002) showed in his study that due to the anthropogenic activities the global, total present-day river input of P

15 has doubled in the last 150 years. In our study the direct anthropogenic influence, such as agricultural input of P into the system, was excluded and should be considered in future studies even though the human impact is projected to decrease until year 3500 (FILIPPELLI, 2008). FILIPPELLI (2008) estimated a rate of 0.03 Tmol P a$^{-1}$ for anthropogenic P delivered to the ocean including fertilization, deforestation and soil loss as well as sewage in year 3000. In comparison to simulated maximum weathering values of 1.09 Tmol P a$^{-1}$ until year 3005 simulated here, the anthropogenic impact seems to be small.

20 **5 Conclusions**

This study constitutes a first approach to estimate the potential impact of changes in the marine P cycle on the expansion of global ocean OMZs under global warming on millennial time scales. Model simulations show that the warming-induced increase in terrestrial weathering (see Fig. 3b) leads to an increase in marine P inventory (see Fig. 2b) resulting in an intensification of the biological pump, corroborating the findings by TSANDEV (2011). As a consequence, oxygen

25 consumption as well as the volume of OMZs increase in our simulations by a factor of 12 over the next millennium (see Fig 2d and 5).

The feedback involving redox-sensitive benthic P fluxes - where the expansion of OMZs leads to an increase in benthic release of P (see Fig. 2c), which in turn enhances biological production and subsequent oxygen consumption (WALLMANN 2010) - has only a limited relevance for the expansion of OMZs in this study. Instead, a negative weathering feedback, by

30 which an increase in alkalinity reduces ocean pCO$_2$ and enhanced P supply leads to an enhanced biological carbon pump, puts some limits to the warming and, eventually, the expansion of the OMZs. The ocean does not become anoxic in our model within the first 10,000 years of simulation, even though the P inventory still increases linearly. We can therefore





conclude that, based on the parameterizations used in this study, the P weathering feedback outcompetes the benthic P feedback on millennial timescales. Although the ocean does not become anoxic in our simulations, the benthic P-release feedback may have played a major role in past oceanic anoxic events. An increase in shelf areas due to higher sea levels, such as during the Cretaceous, would have led to a more powerful benthic P-release feedback. Whether this different

bathymetry alone could result in a dominating benthic release feedback needs to be investigated in future studies.

**Acknowledgments**

This work is a contribution to the Sonderforschungsbereich (SFB) 754 "Climate-Biogeochemical Interactions in the Tropical Ocean". We thank M. Eby for his excellent help with the UVic ESCM. KJM is thankful for a UNSW Science Silverstar Award.

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

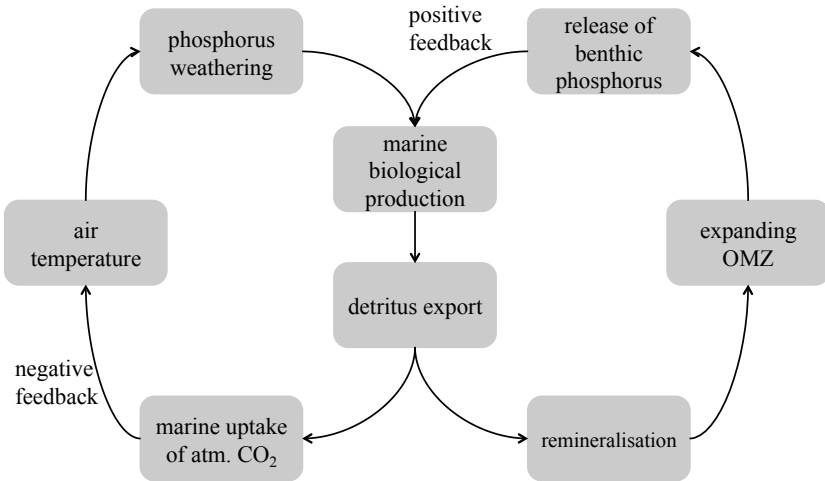

**Figure 1: Possible feedbacks in the global phosphorus cycle under climate warming conditions.**

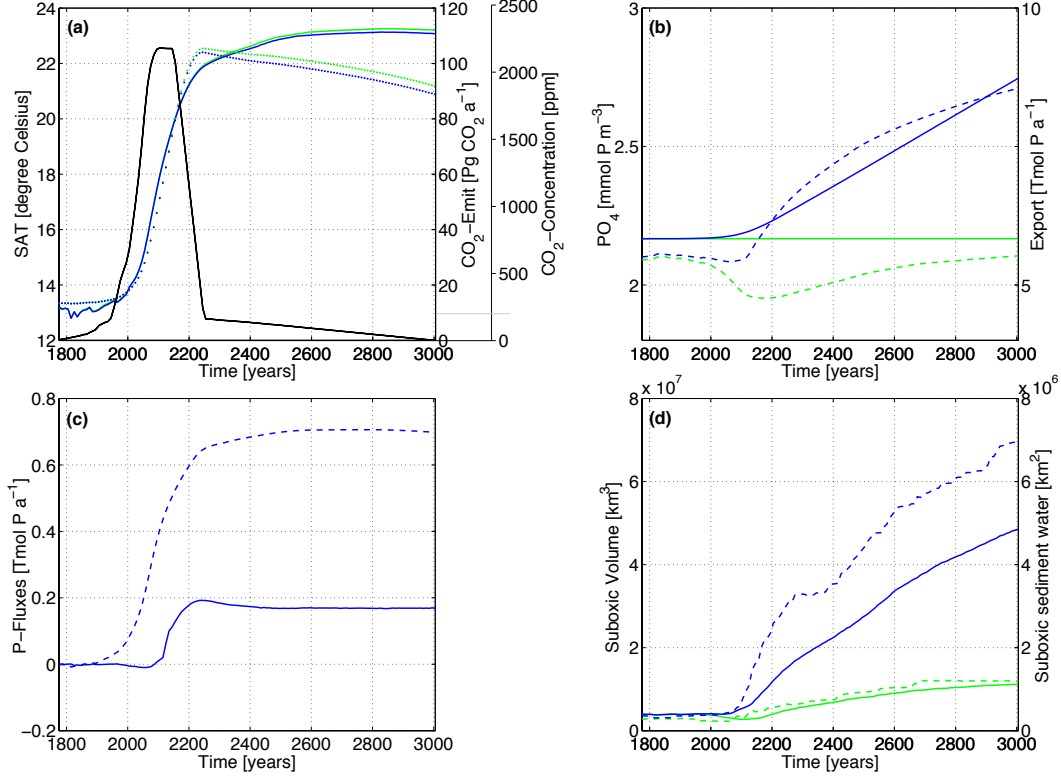

**Figure 2:** Global and annual mean time series of (a) surface air temperature in degree Celsius (solid lines), CO₂-emissions in Pg CO₂ a⁻¹ (black solid line (for both simulations)) and CO₂-concentration in ppm (dotted lines); (b) phosphorus concentration in mmol P m⁻³ (solid lines) and export rate in Tmol P a⁻¹ at 130 m depth (dashed lines); (c) anomalies of phosphorus input via sediment in Tmol P a⁻¹ (solid line) and anomalies of phosphorus weathering input in Tmol P a⁻¹ (dashed line); (d) suboxic volume (<0.005 mol m⁻³) of the ocean in km³ (solid lines) and surface of ocean bottom layer with O₂ concentrations below 0.005 mol m⁻³ (dashed lines). The control simulation (REF) is shown in green, simulation WB in blue.





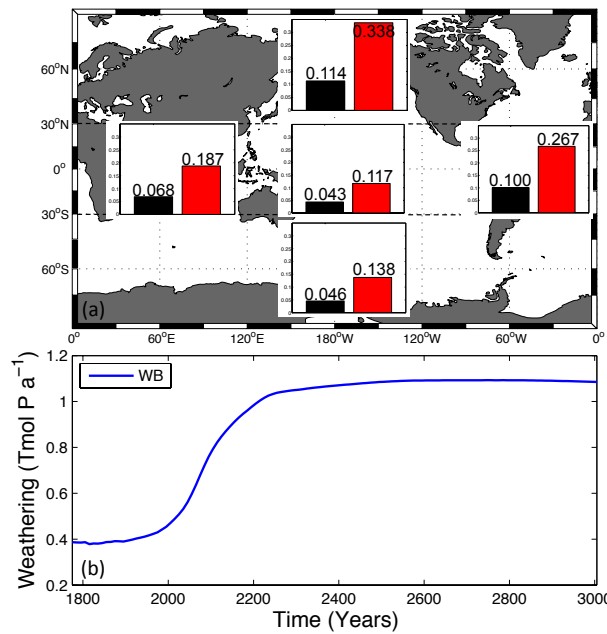

**Figure 3: (a) Phosphorus weathering input (in Tmol a⁻¹) into the Pacific Ocean (middle), Atlantic Ocean (right, middle), Indian Ocean (left, middle), Northern Oceans (=oceans north of 30° N; upper middle) and Southern Ocean (=ocean south of 30° S; lower middle) in 1775 (black bars) and 3005 (red bars). 30° N and 30° S constitute the separating line of calculations. (b): Annual mean averaged phosphorus weathering input (global sum) of 1775 until year 3005.**

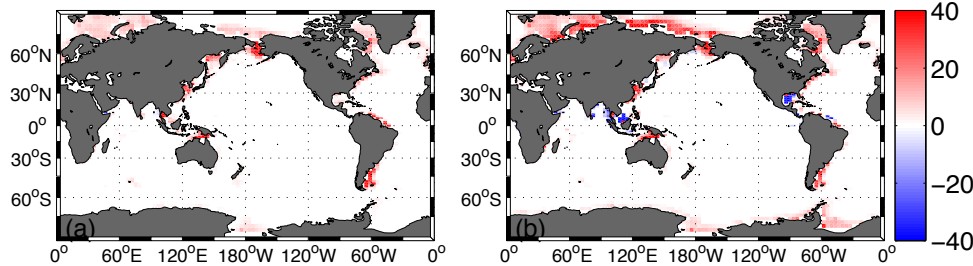

**Figure 4: Difference (year 3005 minus year 1775) in (a) burial and (b) benthic release flux in mmol P m⁻² a⁻¹ for simulation WB.**



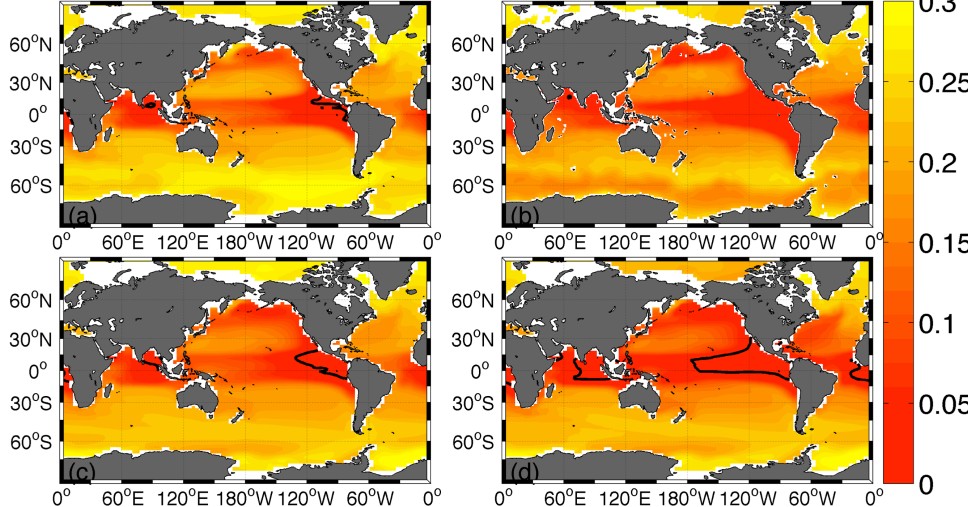

**Figure 5: Oxygen concentration in mol O$_2$ m$^{-3}$ at 300m depth simulated by the (a) control simulation at year 1775 (representative for both REF and WB model runs in year 1775), (b) the World Ocean Atlas in 2009, (c) the control simulation at year 3005 and (d) simulation WB at year 3005. The black isoclines at 0.005 mol m$^{-3}$ highlight the oxygen minimum zones (OMZs).**

