# Peer review of "A model study of warming-induced phosphorus-oxygen feedbacks in open-ocean oxygen minimum zones on millennial timescales"

_Earth System Dynamics, 2016_

## Referee Comment (RC1) · Anonymous Referee #1 · 11 Dec 2016

General comments

This is an interesting paper on coupled phosphorus and oxygen dynamics in the modern ocean essentially showing that both weathering and, to a lesser extent, a phosphorus-oxygen feedback, can contribute to expanding anoxia on a time scale of 1000 kyrs. While the results are important and definitely deserve publication, there are a number of issues that I recommend the authors consider in a revision, as detailed below.

Major issues:

1. The context could be more clearly presented. The point is that you wouldn't really

expect a major impact of P recycling on global ocean biogeochemistry (apart from the coastal zone) on time scales of 1000 kyrs given the relatively long residence time of P in the ocean. In this paper, changes in the modern day ocean under climate change on time scales of 1000 years are repeatedly even compared directly to changes in the Cretaceous Ocean that acted on time scales of 100.000 years and more in an ocean with a very different configuration (other paleogeography, higher sealevel, larger coastal zone). This direct comparison is not recommended: the temporal and spatial scales were just too different. More careful phrasing is thus needed. Furthermore, various recent studies of Cretaceous biogeochemistry suggest that both weathering and recycling of P were needed to sustain the oceanic anoxia (e.g. see the work of Ruvalcaba et al. published in BG in 2014 and the work of Monteiro in GBC).

2. The referencing needs more attention. There are three issues: results of quite some key papers are missing (e.g. relevant to river P fluxes, modeling for the Cretaceous, etc., see below), not all references in the text are in the reference list (see below) and it's strange to cite a thesis when the same work has already been published in the peer-reviewed literature. (work of Tsandev).

3. The writing style can be improved. There are words missing and there are several awkward and/or unclear formulations.

4. Many key issues, such as the role of the coastal zone (in what detail is it included in this model), the bathymetry, the role of anthropogenic fluxes of nutrients (are they considered?) etc. are discussed only at the end of the paper in a section "uncertainties". It would be much better to address these issues up front in the introduction and/or as assumptions in the model description sections.

5. The river fluxes assumed in the model are critical to the results but the river fluxes from the literature discussed seem to be selected rather arbitrarily. I miss references to the work of Ruttenberg (2004; Treatise of Geochemistry), for example. Further details are provided below.

Detailed comments

Page 1. Line 9: replace "reduced solubility" by "reduced oxygen solubility" Page 1. Line 13: replace "Higher availability in phosphorus" by "A higher availability of phosphorus" Page 1. Line 16: the last part of the sentence is repetitive and can be removed, i.e. "which in turn, got there through weathering". Page 1. Lines 19-21. The use of "although" in this sentence suggests that the finding here for the OMZ somehow contrasts with the findings described for the Cretaceous. However, the direct comparison of the impact of the benthic phosphorus release feedback on the size of the OMZ over the coming 1000 years to the feedback on anoxia in the Cretaceous ocean is not appropriate. This is because the relevant processes in the Cretaceous ocean acted on time scales of more than 100,000 years, i.e. there is a 3-order of magnitude difference in time scale. Page 1. Line 21. Previous studies do not "assume" that the P feedback "was the main factor for anoxic conditions during Cretaceous period". They show that both increased river inputs and enhanced regeneration of P likely played a role.

Page 2 Lines 3-4. Sentence is too long. Please break up into two sentences. Line 4. Stramma et al. (2012) and Cabre et al. (2015) are not in the reference list. Line 11. A reference to Tyrell (1999; Nature) would be appropriate here. Line 13. Change to "is river input" Line 18. Specify that you are using the pre-anthropogenic flux of P from rivers. Lines 18-20. Why are the data of Ruttenberg (2004; Treatise of Geochemistry) not used here? That is the most comprehensive summary of reactive and total P fluxes in the marine environment, including river input. Importantly, she reports higher fluxes of P to the marine environment.

Page 3 Lines 8-10. Add references for these statements. Line 11. Change "in previous studies" by "in a previous study". Lines 17-19. Rephrase. It is not correct to refer to Cretaceous studies as "Other studies on OMZs" since in many cases there was not an OMZ but the water column was anoxic to the seafloor also in the deep sea. Line 23. The Cretaceous work by Tsandev et al. was published in EPSL in 2009 – that paper should be referenced, not the PhD-thesis. Line 23. I miss a reference to the modeling

paper of Ruvalcaba et al. (2014; Biogeosciences) here.

Page 4. Section 2.1. I miss details on the bathymetry and how the model deals with coastal processes here. Line 17. Change to "the following equations"

Page 5. How do P burial fluxes calculated in this manner compare to actual P burial fluxes in the ocean per m2 and per time?

Page 6. Line 4. So is burial of P also excluded in this simulation? If so, this should be mentioned specifically ("benthic fluxes" is not generally assumed to refer to burial). Page 6. Line 5. And burial of P? (see previous point). Page 6. Line 5. Change to "was performed" Page 6. Line 6. What about anthropogenic inputs of P? Page 6. Line 19 and further. Here, it is important to distinguish between burial rates for the open ocean and coastal zone. Palastanga et al. used a coarse resolution model that did not resolve the coastal zone and the burial flux thus refers to the open ocean.

Page 6. Note total fluxes of P to the ocean in the published have also been summarized by Ruttenberg (2004; Treatise of Geochemistry) and Slomp (2011; Treatise on Coastal and Estuarine Science) with estimates ranging from 0.258 to 0.92 Tmol yr-1. Part of this total P can indeed by mobilized (i.e. become soluble) in the coastal zone and it is well-known that river fluxes of dissolved P fluxes to the ocean thus underestimate P inputs. Thus there are significantly more data available than suggested here.

Page 7. Lines 1-4: where is the P buried in the current model, i.e. how much is buried in the coastal zone and how much is buried in the open ocean? Again, comparisons should be done carefully: the Palastanga et al. estimates refer to the open ocean because the coastal zone is not well-resolved.

Page 7. Lines 16-17. I would remove this here; because you are pointing forward and are not explaining this fully, it doesn't really fit.

Page 7. Lines 23-24. Where can I see that benthic burial acts as a P sink? Page 8. Lines 11-13. Please show the results for P burial (if you don't want them in the main

[Figure]

paper, add them in a supplement). Page 8. Line 17. Please show these results (see above). Page 8 Line 18: I would suggest to remove "also" Page 8. Line 27: explain :"unlimited" Page 8. Lines 28-30: what about the coastal regions? Page 8: Lines 29-30. Show the result for the 10000 year simulation (e.g. in a supplement) Page 9. Lines 16-17: Based on the residence time of P in the ocean, I wouldn't expect run-away anoxia on a time scale of 1000 years to start with. Page 9. Lines 18-19. Adding "as has been suggested during the Cretaceous" is inappropriate because the processes at the time occurred at a different time scale, thus, a direct comparison should not be made. Note also that we know for certain that parts of the ocean (the proto-North Atlantic) were anoxic in the Cretaceous. Page 9. Line 19. Tsandev et al. 2009; EPSL is the right reference. It would also be logical to discuss the Monteiro et al. and Ruvalcaba-Baroni et al. results here, if you want to discuss modeling results for the Cretaceous. See earlier comment. Page 9: 21-23. I would also add: "present-day paleogeography" because the latter factor also played a role in Cretaceous nutrient cycling. Page 9. The cause of the "unlimited P reservoir" could be better explained, see earlier comment. Page 10. Lines 1-3: It would be better to describe up front in the model how well the coastal zone is resolved and use that information when discussing results and the parameterization (e.g. burial of P and benthic fluxes, see earlier comment). Page 10. Lines 14-19. The potential role of anthropogenic inputs of P to the ocean is better discussed in the model description section, especially because anthropogenic $CO_2$ inputs are considered. Is there not more recent work on anthropogenic inputs of P to the ocean that is relevant to include here, e.g. from the Global NEWS project? (e.g. Harrison et al., Beusen et al. etc.). Page 10. Lines 29-32. This feedback is not explained well.

Figure 2: Please improve the readibility of this figure by adding legends in the panels and/or other markers. It takes a lot of time for the reader to figure out what is what.

———————————————

---

## Referee Comment (RC2) · Anonymous Referee #2 · 12 Dec 2016

Using the UVic Earth System Model the authors describe a feedback loop between expanding oxygen minimum zones (OMZ's) and the availability of dissolved inorganic phosphorus (DIP). A warming climate stimulates weathering processes on land leading to an eutrophication of the oceans. The excess nutrients are taken up by marine phytoplankton which decays due to bacterial decomposition while it sinks out of the euphotic zone, thereby consuming extra oxygen. Increasing benthic oxygen depletion stimulates the redox dependent phosphorus fluxes from sediments, further elevating the concentration levels of DIP, leading to an even larger spread of OMZ's.

General comments: The paper provides an interesting contribution to the actual discussion of the trends of oxygen concentrations under the impact of anthropogenic greenhouse gas emissions. It discusses at the first time the very relevant issue of the threat of an accelerated expansion of OMZ's due to a warming- induced phosphorus-oxygen feedback. The paper is nicely written and I recommend it - subject to minor revisions - for publication in the journal "Earth System Dynamics".

Specific comments: The authors report an increase of ocean net primary production (ONPP) between preindustrial times and year 3005 from 43.8 Tmol P a−1 to 65 Tmol P a−1 even for their reference model run (REF). Usually, global warming is thought to cause a decline in chlorophyll_a concentrations and NPP owing to a strengthening of ocean stratification (see references below).

Gregg W W, Casey N W and McClain C R 2005 Recent trends in global ocean chlorophyll Geophys. Res. Lett. 32 L03606

Boyce D G, Lewis M R and Worm B 2010 Global phytoplankton decline over the past century Nature 466 591–6

The authors should provide a short discussion of how this increase in ONPP, notably in the tropical ocean of their model, can be explained. The volume of OMZ's in UVic under present day conditions is drastically underestimated in comparison with observational data (15.8x106 km3 vs. 102x106 km3). The authors should discuss this flaw more in detail, notably if and how it could influence their conclusions. I would like to see the oxygen concentration map not only in 300 m depth (as shown in Figure 5) but also at depth of 900m.

Technical corrections: In the abstract the sentence ending with: " ... due to increased alkalinity, which, in turn, got there through weathering." sounds awkward. Please rephrase. Page 6 line 19 please replace Tol P a−1 by Tmol P a−1

---

## Author Comment (AC1) · 7 Feb 2017

*Interactive comment on*

**A model study of warming-induced phosphorus-oxygen feedbacks in open-ocean oxygen minimum zones on millennial timescales**

5  Daniela Niemeyer[1], Tronje P. Kemena[1], Katrin J. Meissner[2], Andreas Oschlies[1]

[1]Helmholtz-Zentrum für Ozeanforschung Kiel (GEOMAR), Düsternbrooker Weg 20, 24105 Kiel, Germany
[2]Climate Change Research Centre and ARC Centre of Excellence for Climate System Science, University of New South Wales, Level 4 Mathews Building, Sydney, New South Wales, 2052, Australia

*Correspondence to*: D. Niemeyer (dniemeyer@geomar.de)

15

20

**Responses to Reviewer#1**

We would like to thank the reviewer for their time and expertise. We have replied and acted on all their thoughtful and constructive comments and advice (Reviewer's comments in blue).

General comments

This is an interesting paper on coupled phosphorus and oxygen dynamics in the modern ocean essentially showing that both weathering and, to a lesser extent, a phosphorus-oxygen feedback, can contribute to expanding anoxia on a time scale of 1000 kyrs. While the results are important and definitely deserve publication, there are a number of issues that I recommend

10  the authors consider in a revision, as detailed below.

Major issues:

1. The context could be more clearly presented. The point is that you wouldn't really expect a major impact of P recycling on global ocean biogeochemistry (apart from the coastal zone) on time scales of 1000 kyrs given the relatively long residence

15  time of P in the ocean. In this paper, changes in the modern day ocean under climate change on time scales of 1000 years are repeatedly even compared directly to changes in the Cretaceous Ocean that acted on time scales of 100.000 years and more in an ocean with a very different configuration (other paleogeography, higher sea level, larger coastal zone). This direct comparison is not recommended: the temporal and spatial scales were just too different. More careful phrasing is thus needed. Furthermore, various recent studies of Cretaceous biogeochemistry suggest that both weathering and recycling of P

20  were needed to sustain the oceanic anoxia (e.g. see the work of Ruvalcaba et al. published in BG in 2014 and the work of Monteiro in GBC).

Thank you very much for your advice. We agree with you and will rephrase in the manuscript in a more careful manner and skip the direct comparisons with the Cretaceous.

2. The referencing needs more attention. There are three issues: results of quite some key papers are missing (e.g. relevant to

25  river P fluxes, modeling for the Cretaceous, etc., see below), not all references in the text are in the reference list (see below) and it's strange to cite a thesis when the same work has already been published in the peer-reviewed literature. (work of Tsandev).

Thank you. We will carefully edit the referencing, correct the citation of Tsandev and add some key references (RUTTENBERG (2004); TYRELL (1999); RUVALCABA BARONI ET AL. (2014); SLOMP (2011); RIEBESELL ET AL. (2007)).

30  3. The writing style can be improved. There are words missing and there are several awkward and/or unclear formulations.

We apologize for the writing style of the manuscript. We revised it carefully according to imprecise formulations as well as incomplete sentences. In addition, the revised paper was now proof-read by a native speaker.

4. Many key issues, such as the role of the coastal zone (in what detail is it included in this model), the bathymetry, the role of anthropogenic fluxes of nutrients (are they considered?) etc. are discussed only at the end of the paper in a section

"uncertainties". It would be much better to address these issues up front in the introduction and/or as assumptions in the model description sections.

Thank you. We will add a section about bathymetry and coastal margins in the model description. Although the discussion of these aspects is important, they cannot be fully resolved in this coarse-resolution pilot study. We address these issues

5   together with those of anthropogenic nutrients in section 4 'Uncertainties'. In addition, we now mention in the introduction, that these issues will need more attention in follow-up studies.

5. The river fluxes assumed in the model are critical to the results but the river fluxes from the literature discussed seem to be selected rather arbitrarily. I miss references to the work of Ruttenberg (2004; Treatise of Geochemistry), for example. Further details are provided below.

10   We will include the comprehensive summary of P fluxes by RUTTENBERG (2004). As she reports higher P fluxes to the ocean, our results are in relatively good agreement with fluxes from the literature.

Page 1. Line 9: replace "reduced solubility" by "reduced oxygen solubility"

Corrected.

Page 1. Line 13: replace "Higher availability in phosphorus" by "A higher availability of phosphorus"

15   Corrected.

Page 1. Line 16: the last part of the sentence is repetitive and can be removed, i.e. "which in turn, got there through weathering".

Corrected.

Page 1. Lines 19-21. The use of "although" in this sentence suggests that the finding here for the OMZ somehow contrasts

20   with the findings described for the Cretaceous. However, the direct comparison of the impact of the benthic phosphorus release feedback on the size of the OMZ over the coming 1000 years to the feedback on anoxia in the Cretaceous ocean is not appropriate. This is because the relevant processes in the Cretaceous ocean acted on time scales of more than 100,000 years, i.e. there is a 3-order of magnitude difference in time scale.

Thank you for your careful reading. We corrected the sentence. In the revised manuscript the misleading direct comparison

25   has been removed.

Page 1. Line 21. Previous studies do not "assume" that the P feedback "was the main factor for anoxic conditions during Cretaceous period". They show that both increased river inputs and enhanced regeneration of P likely played a role.

Corrected.

Page 2 Lines 3-4. Sentence is too long. Please break up into two sentences.

30   Corrected.

Line 4. Stramma et al. (2012) and Cabre et al. (2015) are not in the reference list.

Corrected.

Line 11. A reference to Tyrell (1999; Nature) would be appropriate here.

The reference will be added.

*Line 13. Change to "is river input"*

Corrected.

*Line 18. Specify that you are using the pre-anthropogenic flux of P from rivers.*

Thank you for this suggestion to clarify the presentation. We will add this specification.

*Lines 18-20. Why are the data of Ruttenberg (2004; Treatise of Geochemistry) not used here? That is the most comprehensive summary of reactive and total P fluxes in the marine environment, including river input. Importantly, she reports higher fluxes of P to the marine environment.*

Thank you for this suggestion. A comparison to the data by RUTTENBERG (2004) will be added.

*Page 3 Lines 8-10. Add references for these statements.*

We will add the references of SABINE ET AL. (2004), RIEBESELL ET AL. (2007) and ZHANG & CAO (2016) for these statements.

*Line 11. Change "in previous studies" by "in a previous study".*

Corrected.

*Lines 17-19. Rephrase. It is not correct to refer to Cretaceous studies as "Other studies on OMZs" since in many cases there was not an OMZ but the water column was anoxic to the seafloor also in the deep sea.*

We apologize for the imprecise formulation used in this case. We will rephrase as follows:

'Other studies on marine oxygen deficiency focused on the geological past, especially the mid-Cretaceous warm period (120-80 Ma ago) (TSANDEV & SLOMP, 2009; HANDOH & LENTON, 2003; BJERRUM ET AL., 2006; FÖLLMI ET AL., 1996), for which several periods of oceanic oxygen depletion have been inferred from sediment data of black shales (SCHLANGER & JENKYNS, 1976).'

*Line 23. The Cretaceous work by Tsandev et al. was published in EPSL in 2009 – that paper should be referenced, not the PhD-thesis.*

Corrected.

*Line 23. I miss a reference to the modeling paper of Ruvalcaba et al. (2014; Biogeosciences) here.*

Thank you for pointing out this additional reference. We will add the paper of RUVALCABA ET AL. (2014) in the revised manuscript.

*Page 4. Section 2.1. I miss details on the bathymetry and how the model deals with coastal processes here.*

Thank you for your suggestion. We used a sub-grid bathymetry as described in SOMES ET AL. (2013) where smaller-scale (1/5°) features of continental shelves, slopes and other topographical features are included in the computation of biogeochemical fluxes. This favours a relatively precise illustration of benthic burial as well as benthic release, which are – with regard to our study – the most important coastal processes. We will add a description of this approach to the suggested section about bathymetry and coastal processes.

*Line 17. Change to "the following equations"*

Corrected.

 How do P burial fluxes calculated in this manner compare to actual P burial fluxes in the ocean per $m^2$ and per time?

Thank your for your helpful question. We calculated a burial rate of 1.068 mmol $m^{-2}$ $a^{-1}$ in our model. This burial rate is at the upper bound, but still in relatively good agreement with the range reported by RUTTENBERG (2004), who estimated a burial rate of 0.192-0.353 Tmol P $a^{-1}$ (=0.532-0.978 mmol $m^{-2}$ $a^{-1}$). We will add this comparison to our revised manuscript.

5    Page 6. Line 4. So is burial of P also excluded in this simulation? If so, this should be mentioned specifically ("benthic fluxes" is not generally assumed to refer to burial).

We apologize for this misunderstanding; we expressed ourselves in a misleading way. The second simulation includes weathering input, benthic release as well as benthic burial. We used benthic fluxes as an umbrella term for benthic burial and benthic release. In the revised manuscript we will change the wording to:

10    'The second simulation, called WB, includes P weathering as well as benthic burial and release of P but excludes additional anthropogenic input. The spin up was performed by computing the burial and benthic release according to Eq. 6.'

Page 6. Line 5. And burial of P? (see previous point).

Corrected (see previous point).

Page 6. Line 5. Change to "was performed"

15    We think our expression is correct.

'Two model simulations were performed.'

Page 6. Line 6. What about anthropogenic inputs of P?

Thank you for this important question. In comparison to maximum weathering values computed here, the anthropogenic impact is small and was therefore not included in our simulations. This issue is briefly discussed in the discussion section,
20    and we have also now included a statement in the description of the model simulations.

Page 6. Line 19 and further. Here, it is important to distinguish between burial rates for the open ocean and coastal zone. Palastanga et al. used a coarse resolution model that did not resolve the coastal zone and the burial flux thus refers to the open ocean.

Thank you for this important comment. In our study the calculated burial rate for the continental margins (0-200 m) is 0.334
25    Tmol P $a^{-1}$ and for the open ocean (>200 m) 0.046 Tmol P $a^{-1}$. Our estimate is in line with findings presented by RUTTENBERG (2004) and SLOMP (2011). This will be clarified in the revised manuscript.

Page 6. Note total fluxes of P to the ocean in the published have also been summarized by Ruttenberg (2004; Treatise of Geochemistry) and Slomp (2011; Treatise on Coastal and Estuarine Science) with estimates ranging from 0.258 to 0.92 Tmol $yr^{-1}$. Part of this total P can indeed by mobilized (i.e. become soluble) in the coastal zone and it is well-known that
30    river fluxes of dissolved P fluxes to the ocean thus underestimate P inputs. Thus there are significantly more data available than suggested here.

Thank you for your advice. We will add the suggested references in the revised manuscript.

Page 7. Lines 1-4: where is the P buried in the current model, i.e. how much is buried in the coastal zone and how much is buried in the open ocean? Again, comparisons should be done carefully: the Palastanga et al. estimates refer to the open ocean because the coastal zone is not well-resolved.

Thank your for your comment. We agree that a distinction between coastal margin and open ocean is important with regard to benthic burial and benthic release. We will include a comparison with the data compilation by RUTTENBERG (2004) and calculate the benthic release as well as benthic burial in UVic ESCM for continental margins and for the open ocean to compare them with previous studies.

Page 7. Lines 16-17. I would remove this here; because you are pointing forward and are not explaining this fully, it doesn't really fit.

Corrected.

Page 7. Lines 23-24. Where can I see that benthic burial acts as a P sink?

The marine phosphorus inventory in our model is determined by continental inputs through P weathering as the only source and burial at the seafloor as the only the sink of P in the ocean. In order to illustrate this further we will add a figure regarding benthic burial in the supplement (see Figure S 1 and below).

Page 8. Lines 11-13. Please show the results for P burial (if you don't want them in the main paper, add them in a supplement).

We will add P burial results in the supplement (see Figure S 1 and below).

Please see our supplement, where we have now added a figure showing benthic release (see Figure S 1 and below).

Corrected.

We apologise for this imprecise formulation. The benthic release is calculated by subtracting the benthic burial from the particulate organic P rain ratio (see Equation 7). In our simulations the rain ratio shows a strong increase, while the benthic burial is decreasing resulting in an increase of benthic release. Of course, benthic release is limited by the particulate organic P rain ratio. We have removed the term 'unlimited' in the new version.

We agree that the coastal margins are very important. From year 1775 until 10,000 the suboxic volume in the coastal margins increases by about a factor 50 ($O_{2\_1775} = 0.12\%$ of total coastal region; $O_{2\_10000} = 5.57\%$) in the WB-simulation. A maximum increase of 7.29% is simulated from 4785 until 4805, which is still too low for widespread anoxia in coastal regions.

Page 8: Lines 29-30. Show the result for the 10000-year simulation (e.g. in a supplement)

We will add results for the 10,000-year simulation to the supplement (see Figure S 1 and S 3 and below).

Page 9. Lines 16-17: Based on the residence time of P in the ocean, I wouldn't expect run-away anoxia on a time scale of 1000 years to start with.

Thank you for your suggestion. We will reformulate this part.

Page 9. Lines 18-19. Adding "as has been suggested during the Cretaceous" is inappropriate because the processes at the time occurred at a different time scale, thus, a direct comparison should not be made. Note also that we know for certain that parts of the ocean (the proto-North Atlantic) were anoxic in the Cretaceous.

We apologise for the inappropriate statement. As already mentioned above we will skip the direct comparison to the Cretaceous and reformulate this section more carefully.

Page 9. Line 19. Tsandev et al. 2009; EPSL is the right reference. It would also be logical to discuss the Monteiro et al. and Ruvalcaba- Baroni et al. results here, if you want to discuss modeling results for the Cretaceous. See earlier comment.

Thank you for your advice. Instead of discussing modelling results for the Cretaceous, we decided to keep the focus on our own results. Therefore we removed that paragraph.

Page 9: 21-23. I would also add: "present-day paleogeography" because the latter factor also played a role in Cretaceous nutrient cycling.

Thank you for your careful reading. We changed the expression to present-day geography.

Page 9. The cause of the "unlimited P reservoir" could be better explained, see earlier comment.

Please see comment above.

Page 10. Lines 1-3: It would be better to describe up front in the model how well the coastal zone is resolved and use that information when discussing results and the parameterization (e.g. burial of P and benthic fluxes, see earlier comment).

We agree with the reviewer. As already mentioned earlier we now add a short discussion of benthic burial and release in section 'Uncertainties' as well as a description in section 2.1 'UVic Model'.

Page 10. Lines 14-19. The potential role of anthropogenic inputs of P to the ocean is better discussed in the model description section, especially because anthropogenic $CO_2$ inputs are considered. Is there not more recent work on anthropogenic inputs of P to the ocean that is relevant to include here, e.g. from the Global NEWS project? (e.g. Harrison et al., Beusen et al. etc.).

We apologise for this misunderstanding. In our simulation we assume only enhanced weathering rates under climate change conditions, thus we exclude additional anthropogenic inputs into the ocean. We will clarify this in the revised version of the manuscript.

Page 10. Lines 29-32. This feedback is not explained well.

Thank you for pointing this out. We assume an increasing P weathering rate, which enhances the ONPP and the detritus export into the deep ocean. This in turn can lead to an increase in marine uptake of atmospheric $CO_2$, which will impact surface air temperature through a negative feedback loop resulting in a decrease in continental P weathering input. We will clarify the explanation of the feedback in the introduction.

Figure 2: Please improve the readibility of this figure by adding legends in the panels and/or other markers. It takes a lot of time for the reader to figure out what is what.

This is a very good idea. We will improve the readability of the mentioned figure by adding legends in the panels (see Figure 2 in revised manuscript and below).

Figures:

[Figure]

10     **Figure S 1: Global mean and annual mean time series of phosphorus burial (blue solid line; left), phosphorus release (blue dashed line; left) and oxygen (blue solid line; right) for simulation WB until year 10,000.**

[Figure]

**Figure S 3: Oxygen concentration in mol O$_2$ m$^{-3}$ at year 10,000 simulated by the (a) control simulation at 300m depth, (b) and 900m depth, (c) simulation WB at 300m depth and (d) simulation WB at 900 m depth. The black isoclines at 0.005 mol m$^{-3}$ highlight the oxygen minimum zones (OMZs).**

[Figure]

**Figure 2: Global and annual mean time series of (a) Surface Air Temperature in degree Celsius (solid lines), CO₂-emissions in Pg CO₂ a⁻¹ (black solid line (for both simulations)) and CO₂-concentration in ppm (dashed lines); (b) phosphorus concentration in mmol P m⁻³ (solid lines) and export rate in Tmol P a⁻¹ at 130 m depth (dashed lines); (c) anomalies of phosphorus input via sediment in Tmol P a⁻¹ (solid line) and anomalies of phosphorus weathering input in Tmol P a⁻¹ (dashed line); (d) suboxic volume (<0.005 mol m⁻³) of the ocean in km³ (solid lines) and surface of ocean bottom layer with O₂ concentrations below 0.005 mol m⁻³ (dashed lines). The control simulation (REF) is shown in green, simulation WB in blue.**

---

## Author Comment (AC2) · 7 Feb 2017

General comments:
The paper provides an interesting contribution to the actual discussion of the trends of oxygen concentrations under the impact of anthropogenic green house gas emissions. It discusses at the first time the very relevant issue of the threat of an accelerated expansion of OMZ's due to a warming- induced phosphorus-oxygen feedback. The paper is nicely written and I recommend it - subject to minor revisions - for publication in the journal "Earth System Dynamics".

Specific comments
The authors report an increase of ocean net primary production (ONPP) between preindustrial times and year 3005 from 43.8 Tmol P a$^{-1}$ to 65 Tmol P a$^{-1}$ even for their reference model run (REF). Usually, global warming is thought to cause a decline in chlorophyll_a concentrations and NPP owing to a strengthening of ocean stratification (see references below).
Gregg W W, Casey N W and McClain C R 2005 Recent trends in global ocean chlorophyll Geophys. Res. Lett. 32 L03606
Boyce D G, Lewis M R and Worm B 2010 Global phytoplankton decline over the past century Nature 466 591–6
The authors should provide a short discussion of how this increase in ONPP, notably in the tropical ocean of their model, can be explained.

Thank you for this very important comment. In the revised version of the manuscript we will add a discussion about processes altering ONPP under climate change projection as the correlation between temperature and ONPP is still unclear. GREGG ET AL. (2005) and BOYCE ET AL. (2010) suggest a decrease of ONPP under climate change conditions based on enhanced stratification. However, SARMIENTO ET AL. (2004) argued that the temperature sensitivity of ONPP could be the main cause for an increase in ONPP. TAUCHER & OSCHLIES (2011) also highlighted in their study with the UVic-ESCM that simulated future changes in ONPP are very sensitive to the assumed temperature effects on metabolic rates. Furthermore, KVALE ET AL. (2015) found a near-linear relationship between both parameters. Consequently, it can be assumed that the direction of ONPP change depends on the strength of temperature effect versus stratification effect. In our model the direct

effect of temperature on metabolic rates overcompensates the stratification effect and thus leads to a net increase in ONPP under global warming.

The volume of OMZ's in UVic under present day conditions is drastically underestimated in comparison with observational data ($15.8 \times 10^6$ km$^3$ vs. $102 \times 10^6$ km$^3$). The authors should discuss this flaw more in detail, notably if and how it could influence their conclusions. I would like to see the oxygen concentration map not only in 300 m depth (as shown in Figure 5) but also at depth of 900 m.

Because of still relatively poor data coverage and heavy reliance on inter- and extrapolation routines, the data-based estimates for the volumes of OMZs vary widely and the definitions are not yet uniform. Using 20 mmol m$^{-3}$ as threshold for OMZs and comparing our results (WB$_{2005}$ = $1.58 \times 10^7$ km$^3$) with results of the WOA (WOA$_{2005}$=$4.12 \times 10^5$ km$^3$), PAULMIER ET AL. (Paulmier$_{2009}$ = $102 \times 106 \pm 15 \times 10^6$ km$^3$) or BIANCHI ET AL. (Bianchi$_{2012}$ = $2.28$-$2.78 \times 10^6$ km$^3$) it seems that the volume of OMZs are difficult to validate. Our result is within the wide range of these estimates. Looking at the feedbacks described in the manuscript, the suboxic sediment area is more important for the benthic release feedback than the suboxic volume in the water column, which does not necessarily reach the seafloor. In our study we calculated a suboxic sediment area of WB$_{2005}$ = $3.8 \times 10^5$ km$^2$, which fits well with data of the WOA$_{2005}$ = $2.48 \times 10^5$ km$^2$. We will add this discussion to our revised manuscript.

We have also added an oxygen concentration map at a depth of 900 m to the supplement (see Figure S 2 and below).

Technical corrections

In the abstract the sentence ending with: " ... due to increased alkalinity, which, in turn, got there through weathering." sounds awkward. Please rephrase.

Corrected.

Page 6 line 19 please replace Tol P a$^{-1}$ by Tmol P a$^{-1}$

Corrected.

Figures:

[Figure]

**Figure S 2: Oxygen concentration in mol O$_2$ m$^{-3}$ at 900m depth simulated by the (a) control simulation at year 1775 (representative for both REF and WB model runs in year 1775), (b) the World Ocean Atlas in 2009, (c) the control simulation at year 3005 and (d) simulation WB at year 3005. The black isoclines at 0.005 mol m$^{-3}$ highlight the oxygen minimum zones (OMZs).**

5